# TimeKAN: A Transparent KAN-Based Approach for Multivariate Time Series Forecasting

## Abstract

In recent years, numerous deep models have been proposed for the forecasting of multivariate time series (MTS). Transformer-based models show significant potential due to their ability to capture long-term dependencies. However, existing models based on Multi-Layer Perceptrons (MLPs) or Transformers often suffer from a lack of interpretability due to their large parameter sizes, which can be problematic in many real-world applications. To address this issue, we propose **TimeKAN**, a model based on **Kolmogorov-Arnold Networks**. The KAN model offers two key advantages: **(1)** it achieves accuracy comparable to MLPs with significantly fewer parameters, and **(2)** its parameters can be symbolized, which makes it possible to re-interpret the meaning of the parameters. Additionally, instead of the usual attention mechanisms, we designed a **M**ulti-**S**cale **P**atching **(MSP)** module for MTS that allows for a more flexible and simple multi-patching and effectively extracts both temporal and cross-dimensional features. By leveraging this strategy along with KAN, TimeKAN constructs a hierarchical structure capable of utilizing information across different scales, leading to highly accurate predictions. Extensive experiments on six real-world datasets demonstrate that TimeKAN outperforms state-of-the-art (SOTA) methods in terms of predictive performance. Furthermore, we interpret TimeKAN by visualizing its learning process for extracting symbolized features, opening the black box, and revealing meaningful patterns within the time series.

## 1 Introduction

In many real-world scenarios, time series forecasting stands as a highly rewarding endeavor, offering insights that approach the realm of prescience. Multivariate Time Series (MTS) forecasting, a subset of this field, deals with predicting future values based on historical data where each observation contains multiple variables, forming a "channel" dimension. The power of MTS forecasting lies in its ability to enhance decision-making across various domains, including weather (Angryk et al., 2020), electricity (Wu et al., 2021), economic and finance (Loaiza-Maya & Smith, 2020), etc. Recent advancements in deep learning have significantly improved MTS forecasting performance, with methods such as multi-layer perceptrons (MLPs) (Borovykh et al., 2017), recurrent neural networks (RNNs) (Bai et al., 2018; Liu et al., 2020) and transformer-based models (Li et al., 2019).

Although deep learning-based models have made notable progress in time series forecasting, there are still several challenges. Firstly, these methods may not fully capture the complex cross-time and cross-dimension dependencies present in MTS data. While attention mechanisms and Transformer-based models (Chen et al., 2022; Zeng et al., 2023) have improved the ability to capture long-term temporal dependency (cross-time dependency), they often come with increased computational complexity (Liu et al., 2022) and may still fall short in extracting cross-dimension dependencies present in MTS (Zhang & Yan, 2023). Furthermore, the Universal Approximation Theorem (UAT), the mathematical bedrock of most mainstream forecasting models, fails to provide guarantees on the required network size (depth and width) for approximating a given continuous function with specific accuracy (Lu et al., 2021). This limitation, coupled with UAT's ability to only approximate rather than represent, poses a significant hurdle for time series forecasting. Finally, the prediction mechanism of existing models is black-box, resulting in a lack of interpretability. These nontransparent methods are suspected of being suitable for tasks that require a low tolerance for errors, such as medicine, law, and finance.

In this paper, we propose **TimeKAN**, a novel model that leverages Kolmogorov–Arnold Networks (KAN) (Liu et al., 2024) instead of traditional MLP architectures. KAN is a concise neural network architecture inspired by the Kolmogorov-Arnold representation theorem (KART) (Kolmogonov, 1957; 1961). On one hand, KART demonstrates that a multivariate continuous function can be expressed as a combination of a finite number of univariate continuous functions. This theorem links network size to input shape within the context of representation. On the other hand, KAN provides a pruning strategy that simplifies the trained network into a set of symbolic functions, allowing for the analysis of specific modules' mechanisms and greatly enhancing interpretability.

Besides using the KAN architecture, TimeKAN is designed to capture multi-period dependencies between multiple variates through a multi-patching strategy. Unlike single-patching (Zhong et al., 2023) or other attention mechanisms, multi-patching divides the time series data into multi-period patches, enabling the model to focus on a broader spectrum of temporal patterns (e.g. trend and periodicity) in different periods of the data. Moreover, we demonstrate TimeKAN's transparency by symbolizing and visualizing learned meaningful temporal features, as shown in Figure 1, throughout the training process. This transparency is crucial for a wide range of applications where understanding the model's decision-making process is as important as the accuracy of its predictions. **Our main contributions are as follows:**

- We introduce TimeKAN, a KAN-based model that outperforms existing methods, achieving state-of-the-art (SOTA) results on several benchmark datasets. By utilizing kernel-based attention, TimeKAN effectively captures the nonlinear relationships in multivariate time series data.
- We demonstrate that multi-patching significantly improves the model's ability to extract relevant features compared to single-patching or traditional attention mechanisms. Multi-patching allows the model to concurrently analyze multiple segments of the data, enhancing its ability to capture both local and global patterns.
- We employ symbolization techniques to represent the learned features and visualize the training process, addressing the interpretability challenge. This transparency aids in validating the model's performance and understanding the features that contribute to its predictions.

Our code and data are open sourced anonymously at `https://anonymous.4open.science/r/TimeKAN-1BCD/`.

## 2 RELATED WORKS

**Multivariate Time Series Forecasting.** Time series forecasting aims to predict future observations based on historical data. Traditional statistical modeling methods, such as ARIMA (Box & Jenkins, 1968) exponential smoothing (Hyndman et al., 2008) and its variants (Li et al., 2022), have been reliable workhorses for this task. In the realm of deep learning, various architectures have been employed to model different dependencies in time series data. Graph Neural Networks (GNNs) capture spatial dependencies in correlated time series (Jin et al., 2024; Zhao et al., 2023; Miao et al., 2024), while Recurrent Neural Networks (RNNs) are adept at modeling temporal dependencies (Kieu et al., 2022). Models like DeepAR (Rangapuram et al., 2018) combine RNNs with autoregressive methods for short-term forecasting. TimesNet (Wu et al., 2023) transforms one-dimensional time series into two-dimensional representations to capture multi-period features using convolution and shows good performance on four datasets.

Recently, Transformer models have garnered significant attention in multivariate time series forecasting (Wu et al., 2020; Zerveas et al., 2021; Wen et al., 2022). Informer (Zhou et al., 2021) introduces prob-sparse self-attention to select important keys, and Triformer (Cirstea et al., 2022) employs a triangular architecture to reduce complexity. Autoformer (Wu et al., 2021) replaces self-attention with auto-correlation mechanisms to model temporal dynamics, while FEDformer (Zhou et al., 2022) utilizes Fourier transformations from a frequency perspective. However, some studies have raised concerns about the effectiveness of Transformers in this domain, as simple linear models have proven to be effective or even outperform previous Transformer-based models Li et al. (2022); Zeng et al. (2023). Nonetheless, PatchTST (Nie et al., 2022) enhances performance by employing patching and channel independence within Transformers, indicating that the Transformer architecture still holds potential with proper adaptations in time series forecasting.

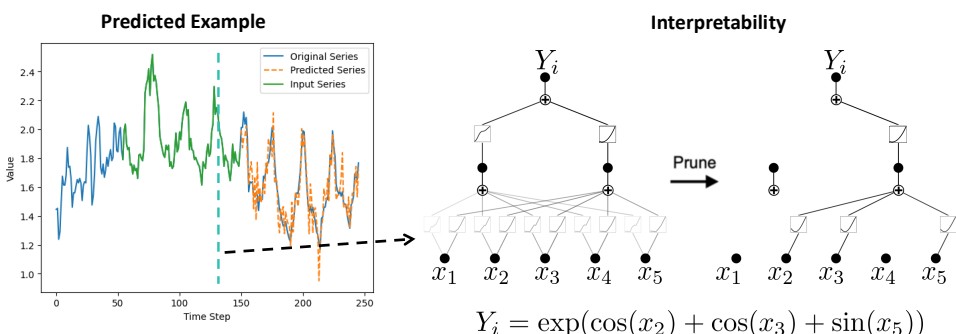

$$Y_i = \exp(\cos(x_2) + \cos(x_3) + \sin(x_5))$$

Figure 1: Predictive and interpretable capabilities of KAN in time series. The left panel illustrates the fidelity of the KAN-based model predictions against the original series (performed as yellow lines), underscoring its predictive accuracy. The light blue vertical lines represent the input series in single patch after it has been split into multiple patches. The right panel elucidates the fitting of mathematical expressions to learnable activation functions, thereby enhancing transparency by elucidating the influence of past observations on future predictions.

**Multi-scale Decomposition for Time Seriess.** N-BEATS (Oreshkin et al., 2019), N-HiTS (Challu et al., 2023), and ETSformer (Woo et al., 2022) apply decomposition to deep learning and show satisfactory results in time series forecasting. However, N-BEATS and N-HiTS do not consider the inter-channel correlation, which has been shown critical in MTS analysis tasks. In addition, they are based on plain MLP on the temporal dimension while ETSformer is based on self-attention for temporal modeling, all of which do not take into account the sub-series level features. To capture intra- and inter-patch variations and channel-wise correlations, MSD-Mixer (Zhong et al., 2023) consider the residual of the decomposition, which employs MLPs handle the multi-scale temporal patterns and multivariate dependencies. Meanwhile, Pyraformer (Liu et al., 2022) introduces a pyramid attention to extract features at different temporal resolutions. Scaleformer (Shabani et al., 2023) proposes a multi-scale framework, and the need to allocate a predictive model at different temporal resolutions results in higher model complexity. (Chen et al., 2024) These works have encouraged us to build a flexible and adaptive multi-scale patching module rather than fixed-size ones.

**KAN-based models for MTS.** Recently, KAN has been introduced into time series forecasting to address challenges such as unclear relations between network sizes and fitting capabilities, and the lack of interpretability in deep learning models. Han et al. (2024) proposed the Reversible Mixture of KAN Experts (RMoK) model, which employs a mixture-of-experts structure to assign variables to KAN experts, achieving state-of-the-art performance on several real-world datasets. Similarly, Xu et al. (2024) developed Temporal KAN (T-KAN) and Multivariate Temporal KAN (MT-KAN), focusing on detecting concept drift and enhancing interpretability through symbolic regression. These studies demonstrate that KAN-based models effectively bridge the gap between predictive power and interpretability in time series forecasting. However, their proposed KAN-based single-layer model is only a preliminary attempt to introduce KAN into time series forecasting. We hope to further improve the performance and interpretability of the KAN-based model by combining the Multi-Scale property of time series.

## 3 TIMEKAN: INTERPRETABLE KAN-BASED MODEL FOR TIME SERIES

As aforementioned, time series usually exhibit complex patterns and require strong interpretability in real-world applications, especially in areas such as finance and biology. Inspired by the effectiveness of multi-scale modeling and the interpretable capabilities of KAN in MTS showed in Figure 1, the core idea of **TimeKAN** is to introduce an multi-scale decomposition architecture and interpretable process to the underlying network based on KAN. In this section, we begin by presenting an overview of the TimeKAN architecture in Section 3.1. We then delve into the key components of the model, including the incremental decomposition strategy in Section 3.2 and integration of Kolmogorov-Arnold Networks (KAN) from Section 3.6 to Section 3.7.

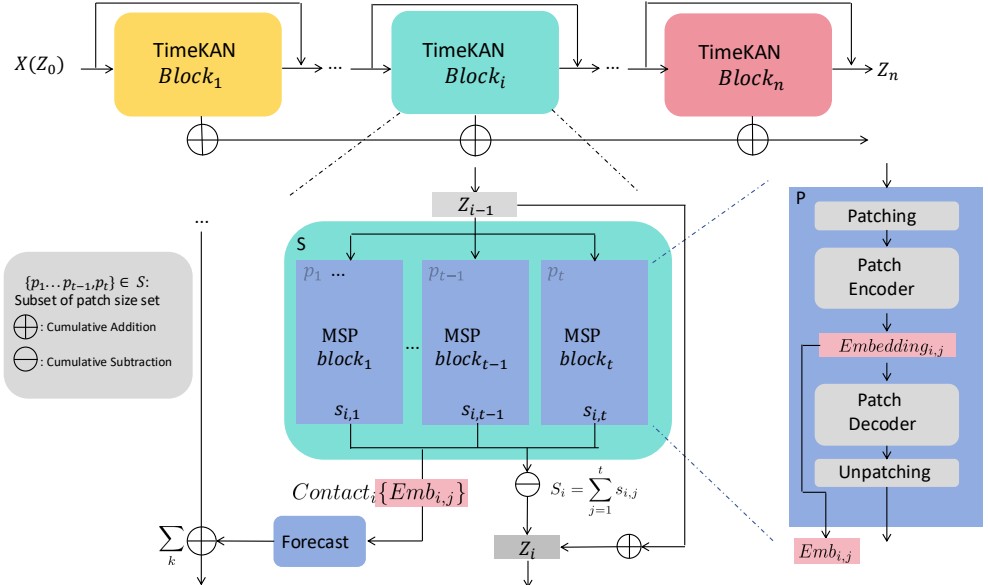

Figure 2: The architecture of TimeKAN. TimeKAN comprises a stack of n layers, and learns to hierarchically decompose the input X into n components $\{S_1, S_2, \ldots, S_n\}$. The number of layers and components n is a hyperparameter in TimeKAN which should be determined according to the properties of the dataset.

## 3.1 ARCHITECTURE OVERVIEW

TimeKAN's architecture, illustrated in Figure 2, is a hierarchical model composed of $n$ residual-connected layers, referred to as *TimeKAN blocks*. Each block contains multiple *Multi-Scale Patching* (MSP) modules designed to capture temporal dependencies at various scales. The input sequence $X \in \mathbb{R}^{T \times D}$, where $T$ is the sequence length and $D$ is the feature dimension, is incrementally decomposed into a set of components $\{S_1, S_2, \ldots, S_n\}$, enabling the model to extract meaningful patterns progressively.

## 3.2 MULTI-SCALE DECOMPOSITION

The TimeKAN model employs an incremental decomposition strategy inspired by residual learning. The process begins with the initialization:

$$Z_0 = X, \tag{1}$$

where $Z_0$ is the initial residual input. Each TimeKAN block aims to model the residual between the current input and the cumulative components extracted so far. This is formulated as:

$$Z_i = Z_{i-1} - S_i, \quad \text{for } i = 1, 2, \ldots, n, \tag{2}$$

where $Z_i \in \mathbb{R}^{T \times D}$ is the residual input to the $i$-th block, and $S_i \in \mathbb{R}^{T \times D}$ is the component extracted by the $i$-th block. This residual learning mechanism enables each block to focus on modeling the discrepancies not captured by preceding blocks.

Each component $S_i$ is further decomposed into multiple sub-components to capture multi-scale temporal dynamics:

$$S_i = \sum_{j=1}^{t} s_{i,j}, \tag{3}$$

where $s_{i,j} \in \mathbb{R}^{T \times D}$ denotes the $j$-th sub-component of the $i$-th block, and $t$ is the number of MSP modules within each block.

Figure 3: Illustration of multi-scale patching in the MSP module. Given an input time series with $C$ channels and $L$ time steps, we transform it into patches of size $p_i$ through the following steps: (1) Pad the beginning of the time series with zeros to ensure that the total length $L$ is divisible by the patch size $p_i$; (2) Segment the padded time series along the temporal dimension into non-overlapping patches using a stride of $p_i$, resulting in $L' = \lfloor L/p_i \rfloor$ patches; (3) Permute the data to introduce a new patch dimension, yielding a tensor of shape $C \times L' \times p_i$.

### 3.3 MULTI-SCALE PATCHING (MSP) MODULES

The MSP modules are designed to capture temporal patterns at various scales by processing the residual input $Z_{i-1}$ through different patch sizes. Each MSP module operates independently to focus on specific temporal resolutions.

**Patch Generation.** Given the residual input $Z_{i-1}$, each MSP module partitions the sequence into patches of size $p_j$, where $j = 1, 2, \ldots, t$. This process enables the model to process segments of the time series at the appropriate temporal resolution. The patches are defined as:

$$Z_{i-1}^{(j)} = \text{Patchify}\left(Z_{i-1}, p_j\right), \tag{4}$$

where $\text{Patchify}(\cdot)$ is a function that segments the sequence into non-overlapping patches of length $p_j$.

**How to Determine Patching Size:** Selecting appropriate patching sizes $p_j$ is crucial for capturing relevant temporal patterns. To determine the optimal patching size, we utilize the Fast Fourier Transform (FFT) on a one-dimensional time series $X \in \mathbb{R}^{T \times C}$ with time length $T$ and channel dimension $C$(Wu et al., 2023). This process allows us to identify the periodicity of the series. For each dimension of the multivariate time series $X$, we compute the amplitude spectrum to identify significant frequencies:

$$A = \frac{1}{D} \sum_{d=1}^{D} |\text{FFT}\left(X_{:,d}\right)|, \tag{5}$$

where $X_{:,d}$ denotes the $d$-th feature over all time steps, and $|\cdot|$ denotes the magnitude of the complex Fourier coefficients. We identify the top $k$ frequencies with the highest amplitudes, $\{f_1, f_2, \ldots, f_k\}$, and compute the corresponding periods:

$$p_j = \left\lfloor \frac{T}{f_j} \right\rfloor, \quad j = 1, 2, \ldots, k. \tag{6}$$

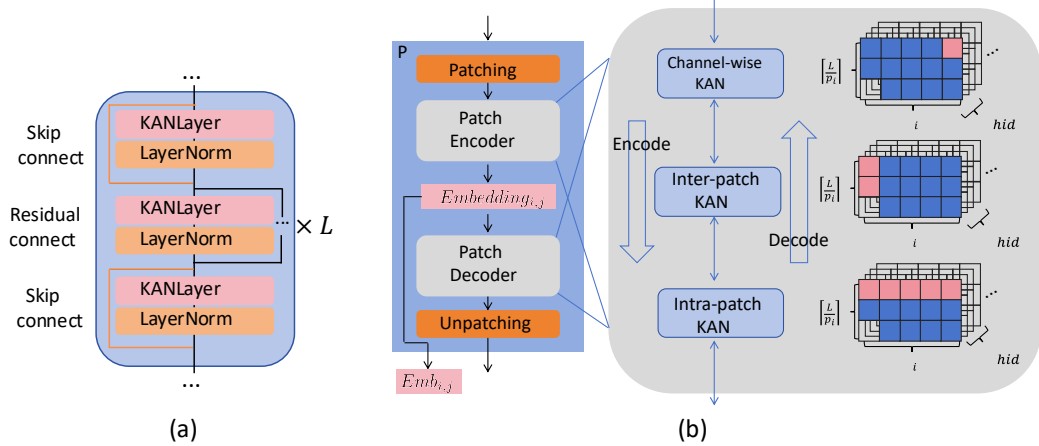

(a)                                              (b)

Figure 4: (a) KAN block. (b) Patch Encoder and Decoder: The Patch Encoder includes a channel-wise KAN block, inter-patch and intra-patch MLP blocks, and a linear layer to create the component representation $\mathbf{E}_i$ from patched $\mathbf{Z}_{i-1}$. The Patch Decoder reverses this order to reconstruct $\mathbf{S}_i$ from $\mathbf{E}_i$.

We denote $\{p_1, p_2, \ldots, p_k\}$ as $P$ for a patching size sequence. For a dataset with $C$ dimensions, we can view it as $C$ one-dimensional time series and get $C$ patching size sequences $\{P_1, P_2, \ldots, P_C\}$. These periods are used as patch sizes for the MSP modules, ensuring that the model is tuned to capture the most significant temporal patterns present in the data.

## 3.4 ENCODER-DECODER STRUCTURE

Inspired by autoencoders with **Encoder** to extract high-level feature embeddings from the input patches and **Decoder** to reconstruct the sub-component $s_{i,j}$ from the embedding, the Patch Encoder and Decoder modules are based exclusively on KANs along different dimensions for feature extraction. We show the design of each multi-layer KAN block in Figure 4 (a), which simply consists of multiple KANLayer and LayerNorm combinations, repeated $L + 2$ times. It includes residual connections that link inputs directly to outputs, enhancing gradient flow, and skip connections that allow information to bypass certain layers, improving efficiency and learning. This module can act on channel wise, inter-patch, and intra-patch dimensions as follows:

- The *channel-wise KAN block* allows communication between different channels, to capture inter-channel correlations.

- The *inter-patch KAN block* allows communication between different patches, to capture global contexts.

- The *intra-patch KAN block* allows communication between different time steps within a patch, to capture sub-series level variations.

Each MSP module employs an encoder-decoder architecture as showed in Figure 4 (b). The encoder for the $j$-th MSP module in block $i$ is defined as $E_{i,j} = \text{Encoder}_{i,j}\left(Z_{i-1}^{(j)}\right)$, where $E_{i,j} \in \mathbb{R}^{T_j \times D'}$ is the embedding output, $T_j = \frac{T}{p_j}$ is the adjusted sequence length based on the patch size, and $D'$ is the embedding dimension. The decoder for the $j$-th MSP module in block $i$ is defined as: $s_{i,j} = \text{Decoder}_{i,j}(E_{i,j})$. This reconstruction allows the model to focus on specific temporal patterns captured by the corresponding encoder.

## 3.5 FORECAST GENERATION

After processing through all TimeKAN blocks, the model aggregates the embeddings from all MSP modules to generate the final forecast.

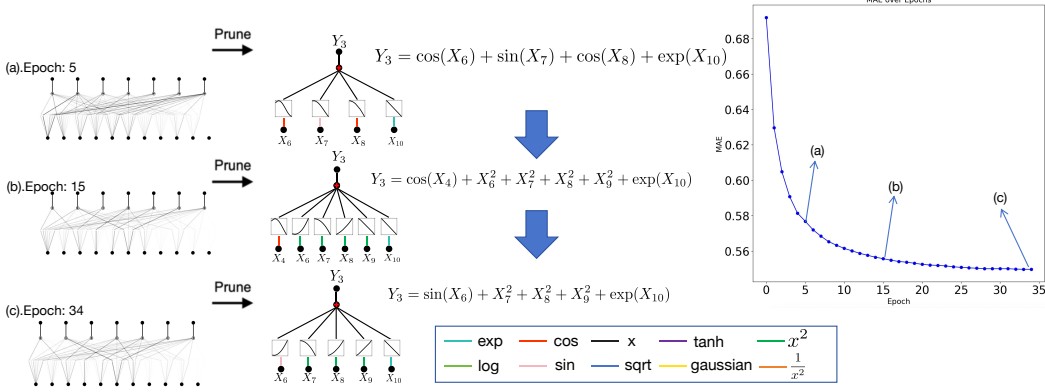

Figure 5: Transparent training process. The figure above depicts an example of how the symbolic representation of our TimeKAN changes with respect to the Val loss during the training of the Channel-wise component of the Encoder at a certain patch scale.

**Embedding Concatenation.** The embeddings from all MSP modules across all blocks are concatenated to form a comprehensive feature representation:

$$E = \text{Concat}\left(E_{i,j} \mid i = 1, 2, \ldots, n; \ j = 1, 2, \ldots, t\right). \tag{7}$$

This aggregated embedding combines multi-scale temporal features extracted throughout the entire model, providing rich information for prediction.

**Prediction Module.** The aggregated embedding $E$ is passed through a prediction module to generate the final forecast $\hat{Y} = \text{Forecast}(E)$.

The prediction module leverages the multi-scale embeddings to produce accurate forecasts for future time steps. It maps the aggregated embedding $E$ to the desired output space, producing the final prediction $\hat{Y} \in \mathbb{R}^{T_{\text{pred}} \times D}$, where $T_{\text{pred}}$ is the prediction horizon.

### 3.6 KOLMOGOROV-ARNOLD NETWORKS (KAN)

At the core of each MSP module lies the *Kolmogorov-Arnold Network* (KAN), a theoretical framework that facilitates the approximation of multivariate continuous functions through a combination of univariate and bivariate function compositions (Kolmogonov, 1957; 1961). By integrating KAN structures within the MSP modules, TimeKAN effectively captures complex temporal dependencies across different scales. A symbolic representation of the training results of any KAN layer inside the patch encoder is visible at a certain patching scale by taking out the TimeKAN block of any layer, which means the time series at that scale.

As shown in Figure 1, we can get $\ln(Y_i) = \cos(x_2) + \cos(x_3) + \sin(x_5)$ by taking the natural logarithm of both side, which is equivalent to the logarithmic rate of return in finance. This equation reveals that the yield $Y_i$ is influenced by the periodic functions $\cos(x_2)$, $\cos(x_3)$, and $\sin(x_5)$. The presence of cosine and sine functions indicates that there are periodic features within the return series. These components capture cyclical patterns or oscillations over the variables $x_2$, $x_3$, and $x_5$, suggesting that the underlying process has inherent periodicity. This insight is crucial for understanding and interpreting the dynamics and regularities in the series, potentially aiding in forecasting or pattern recognition tasks.

### 3.7 SYMBOLIC REGRESSION FOR INTERPRETABILITY

In fact, the statistical model represented by ARIMA is still widely used today because, although the deep learning network approach often achieves good results on the dataset, the statistical model has good interpretability and can be demonstrated for seasonality, trends and other data characteristics. Inspired by this, a transparent training process using symbolic regression is incorporated into TimeKAN to enhance interpretability by fitting mathematical expressions to the learnable activation functions. This approach allows us to generate human-readable models that explain the underlying

patterns in the data. This synergy between KAN and symbolic regression allows us to understand how past observations influence future predictions in a transparent manner.

Figure 5 illustrates this process, showcasing how the combined model yields interpretable representations that reveal the temporal dynamics and dependencies inherent in the time series data.

# 4    EXPERIMENTS

In this section, we rigorously evaluate the performance of our proposed method on several long-term forecasting tasks using diverse real-world time series datasets (please refer to Appendix A.1 for more details). We compare our TimeKAN against various state-of-the-art baseline models, encompassing both MLP-based and Transformer-based architectures. Detailed implementation settings are provided in Appendix A.2 ensure reproducibility and facilitate future research.

## 4.1    COMPARED MODELS

To demonstrate the superiority of our proposed method, we conducted experiments across 27 benchmarks on the aforementioned datasets, comparing our TimeKAN against selected baselines. These baselines include models based on recent MLP and Transformer architectures, as shown below.

Regarding Transformer-based model, **Mamba** (Gu & Dao, 2023) is a forecasting model that achieves promising prediction accuracy while maintaining low computational complexity. **TimesNet** (Wu et al., 2023) proposes a framework based on time-domain two-dimensional variation modeling. **PatchTST** (Nie et al., 2022) divides time series into multiple "patches" and utilizes Transformer to capture the temporal patterns in these patches. **iTransformer** (Liu et al., 2023) focuses on adaptive modeling of time series data by introducing learnable time encoding and dynamic attention mechanisms.

As for the Multi-Layer Perceptron (MLP) based model, **DLinear** (Zeng et al., 2023) decomposes the time series into trend and seasonal components using moving averages and models each component using linear models. **LightTS** (Zhang et al., 2022): introduce a lightweight time series forecasting model that enhances predictive capability through auxiliary sequences and feature selection mechanisms. **MSD-Mixer** (Zhong et al., 2023) utilizes a multi-scale distribution mixing method through a fully connected neural network architecture to capture multi-scale features in time series data.

## 4.2    RESULT ANALYSIS

As shown in the Table 1, our extensive experiments across various datasets and forecasting horizons demonstrate the superior performance of TimeKAN compared to existing models. TimeKAN consistently outperforms or matches the best existing MTS models across most datasets and forecasting horizons, showing particular strength in MTS forecasting. For instance, on ETTh1, TimeKAN shows improvements of up to 3.4% in MSE compared to the previous best model (PatchTST) for the 336-hour forecast. In the Exchange dataset, TimeKAN outperforms other models by 6.9% and 2.2% in MSE, respectively. The model's strong performance extends across various forecasting horizons, from short-term (96 hours) to long-term (720 hours) predictions, demonstrating its versatility and robustness.

TimeKAN's effectiveness is further emphasized when compared to recent state-of-the-art network architectures. It consistently outperforms or shows comparable results to MSD-Mixer, particularly excelling in the ETTm1 and ETTh1 datasets. Against Transformer-based models like iTransformer and TimesNet, TimeKAN demonstrates superior performance, especially in longer forecasting horizons. Notably, TimeKAN also outperforms the recent Mamba model across all datasets and horizons, often by a significant margin. While TimeKAN shows strong performance across most scenarios, there are areas for potential improvement, such as in very long-term financial forecasting where DLinear performs better for the 720-hour forecast in the Exchange dataset. Overall, the experimental results strongly support the effectiveness of TimeKAN in long-term time series forecasting across various domains, demonstrating its robustness and generalizability for diverse scenarios.

Table 1: Performance of long-term forecasting across various time series benchmark datasets. For each dataset, the best-performing results are highlighted in bold, while the second-best results are underlined. Note that the MSE/MAE results have been derived from different parameter settings.

| Model | | TimeKAN (Ours) | | MSD-Mixer (2024) | | iTransformer (2022) | | Mamba (2024) | | LightTS (2022) | | TimesNet (2023) | | PatchTST (2023) | | DLinear (2023) | |
|---|---|---|---|---|---|---|---|---|---|---|---|---|---|---|---|---|---|
| Metric | | MSE | MAE | MSE | MAE | MSE | MAE | MSE | MAE | MSE | MAE | MSE | MAE | MSE | MAE | MSE | MAE |
| ETTm1 | 96 | **0.310** | **0.356** | 0.313 | 0.357 | 0.343 | 0.377 | 0.365 | 0.387 | 0.378 | 0.404 | 0.330 | 0.370 | 0.323 | 0.364 | 0.345 | 0.372 |
| | 192 | **0.350** | 0.384 | 0.351 | **0.380** | 0.380 | 0.394 | 0.434 | 0.419 | 0.418 | 0.430 | 0.387 | 0.398 | 0.366 | 0.387 | 0.382 | 0.391 |
| | 336 | **0.368** | 0.401 | 0.376 | **0.397** | 0.419 | 0.419 | 0.526 | 0.489 | 0.460 | 0.460 | 0.430 | 0.428 | 0.399 | 0.408 | 0.413 | 0.413 |
| | 720 | **0.433** | 0.440 | 0.435 | **0.431** | 0.488 | 0.457 | 0.637 | 0.546 | 0.534 | 0.509 | 0.494 | 0.462 | 0.458 | 0.445 | 0.472 | 0.450 |
| ETTm2 | 96 | 0.176 | 0.266 | **0.174** | **0.263** | 0.185 | 0.270 | 0.206 | 0.282 | 0.228 | 0.324 | 0.188 | 0.269 | 0.183 | 0.270 | 0.193 | 0.291 |
| | 192 | 0.249 | 0.321 | **0.237** | **0.303** | 0.254 | 0.314 | 0.292 | 0.338 | 0.342 | 0.406 | 0.252 | 0.308 | 0.252 | 0.312 | 0.283 | 0.359 |
| | 336 | 0.313 | 0.356 | **0.296** | **0.340** | 0.313 | 0.351 | 0.354 | 0.381 | 0.474 | 0.485 | 0.312 | 0.344 | 0.309 | 0.348 | 0.372 | 0.420 |
| | 720 | 0.410 | 0.418 | **0.397** | **0.403** | 0.412 | 0.405 | 0.545 | 0.472 | 0.818 | 0.654 | 0.430 | 0.411 | 0.411 | 0.445 | 0.835 | 0.654 |
| ETTh1 | 96 | **0.376** | **0.395** | 0.383 | 0.397 | 0.394 | 0.409 | 0.491 | 0.457 | 0.461 | 0.458 | 0.398 | 0.418 | 0.387 | 0.403 | 0.397 | 0.411 |
| | 192 | **0.419** | 0.426 | 0.427 | **0.424** | 0.449 | 0.442 | 0.568 | 0.510 | 0.516 | 0.491 | 0.475 | 0.465 | 0.428 | 0.432 | 0.445 | 0.440 |
| | 336 | **0.449** | **0.450** | 0.472 | 0.452 | 0.491 | 0.465 | 0.526 | 0.488 | 0.564 | 0.519 | 0.526 | 0.490 | 0.465 | 0.455 | 0.487 | 0.465 |
| | 720 | **0.464** | **0.475** | 0.484 | 0.480 | 0.518 | 0.501 | 0.622 | 0.573 | 0.589 | 0.549 | 0.518 | 0.495 | 0.529 | 0.508 | 0.515 | 0.512 |
| ETTh2 | 96 | **0.282** | **0.342** | 0.293 | 0.351 | 0.301 | 0.351 | 0.356 | 0.383 | 0.413 | 0.446 | 0.313 | 0.358 | 0.302 | 0.350 | 0.354 | 0.405 |
| | 192 | 0.365 | **0.391** | **0.363** | 0.394 | 0.379 | 0.399 | 0.448 | 0.442 | 0.522 | 0.507 | 0.473 | 0.453 | 0.372 | 0.399 | 0.482 | 0.479 |
| | 336 | **0.407** | **0.427** | 0.411 | 0.432 | 0.422 | 0.432 | 0.439 | 0.446 | 0.660 | 0.573 | 0.469 | 0.463 | 0.423 | 0.441 | 0.588 | 0.539 |
| | 720 | **0.415** | 0.448 | 0.416 | 0.448 | 0.428 | 0.447 | 0.497 | 0.489 | 0.983 | 0.710 | 0.455 | 0.460 | 0.426 | **0.446** | 0.835 | 0.659 |
| Exchange | 96 | **0.081** | **0.200** | 0.087 | 0.207 | 0.097 | 0.222 | 0.135 | 0.266 | 0.127 | 0.265 | 0.114 | 0.243 | 0.092 | 0.210 | 0.094 | 0.226 |
| | 192 | **0.179** | **0.300** | 0.189 | 0.305 | 0.184 | 0.309 | 0.288 | 0.396 | 0.263 | 0.385 | 0.202 | 0.326 | 0.183 | 0.303 | 0.185 | 0.325 |
| | 336 | 0.351 | 0.424 | 0.357 | 0.427 | **0.329** | 0.417 | 0.681 | 0.621 | 0.538 | 0.552 | 0.379 | 0.447 | 0.330 | **0.416** | 0.330 | 0.437 |
| | 720 | 0.892 | 0.708 | 0.929 | 0.731 | 0.925 | 0.710 | 1.695 | 0.969 | 0.936 | 0.739 | 0.945 | 0.736 | 0.938 | 0.727 | **0.774** | **0.673** |
| Weather | 96 | 0.153 | 0.217 | **0.149** | **0.213** | 0.175 | 0.214 | 0.192 | 0.241 | 0.176 | 0.237 | 0.173 | 0.223 | 0.175 | 0.217 | 0.176 | 0.237 |
| | 192 | **0.201** | 0.267 | 0.202 | 0.267 | 0.224 | 0.256 | 0.255 | 0.293 | 0.219 | 0.279 | 0.224 | 0.265 | 0.225 | 0.261 | 0.219 | 0.279 |
| | 336 | **0.256** | 0.310 | 0.259 | 0.311 | 0.280 | **0.300** | 0.330 | 0.348 | 0.268 | 0.316 | 0.290 | 0.311 | 0.281 | **0.300** | 0.268 | 0.316 |
| | 720 | **0.326** | 0.361 | 0.338 | 0.368 | 0.357 | **0.349** | 0.405 | 0.385 | 0.332 | 0.363 | 0.360 | 0.355 | 0.356 | **0.349** | 0.332 | 0.363 |
| Exg Pro | 96 | **0.177** | **0.259** | 0.180 | **0.259** | 0.188 | 0.267 | 0.269 | 0.327 | 0.227 | 0.309 | 0.273 | 0.323 | 0.176 | 0.260 | 0.226 | 0.308 |
| | 192 | 0.354 | 0.370 | 0.345 | 0.368 | 0.374 | 0.388 | 0.556 | 0.489 | 0.421 | 0.431 | 0.550 | 0.458 | **0.344** | 0.371 | 0.455 | 0.448 |
| | 336 | **0.610** | **0.503** | 0.614 | **0.503** | 0.628 | 0.510 | 1.019 | 0.677 | 0.871 | 0.634 | 0.708 | 0.548 | 0.622 | 0.511 | 0.900 | 0.643 |
| | 720 | - | - | - | - | - | - | - | - | - | - | - | - | - | - | - | - |

## 4.3 ABLATION STUDY

Our research reveals that our TimeKAN's structure, under specific configurations (special cases), is equivalent at the architectural level to previously effective models. To validate the efficacy of our structural design, we conducted ablation studies focusing on various aspects, including the number of MSP layers and TimeKAN layers. Partial results are presented in Table 2 and 3, with additional details provided in Appendix A.5.

To demonstrate that the MSP layer is indeed a crucial component of our TimeKAN, we conducted a carefully designed ablation study on the number of MSP blocks in each TimeKAN block. In this experiment, we varied the number of MSP blocks from 1 to 6 and evaluated the model's performance on our selected benchmark datasets (Some of the results are shown in Table 2.). The results revealed a strong correlation between the number of MSP blocks in TimeKAN and the model's overall performance. This succinctly proves that the MSP layer plays a vital role in our TimeKAN's architecture and effectiveness.

Table 2: Ablation study for the number of MSP layer.

| # MSP Blocks | Metric | 1 | | 2 | | 3 | | 4 | | 5 | |
|---|---|---|---|---|---|---|---|---|---|---|---|
| | | MSE | MAE | MSE | MAE | MSE | MAE | MSE | MAE | MSE | MAE |
| Exchange | 96 | 0.100 | 0.223 | 0.089 | 0.206 | 0.089 | 0.207 | 0.101 | 0.218 | 0.082 | 0.201 |
| | 192 | 0.211 | 0.327 | 0.175 | 0.302 | 0.167 | 0.289 | 0.196 | 0.311 | 0.178 | 0.299 |
| | 336 | 0.434 | 0.474 | 0.405 | 0.462 | 0.303 | 0.399 | 0.381 | 0.448 | 0.349 | 0.431 |

Furthermore, to validate the efficacy of KAN in our work, we conducted a comparative experiment, replacing KAN with MLP while maintaining identical structural parameters. Results demonstrate that KAN outperforms MLP in our model architecture, confirming its effectiveness and justifying its use in our approach. (Some of the results are shown in Table 3.)

Table 3: Comparison of the performance between MLP and KAN i-th the same parameters except for the hidden layer parameters.

| Dataset | Model | 96 | | 192 | | 336 | | 720 | |
|---|---|---|---|---|---|---|---|---|---|
| | | MSE | MAE | MSE | MAE | MSE | MAE | MSE | MAE |
| ETTm1 | MLP | 0.313 | 0.358 | 0.367 | 0.388 | 0.374 | 0.407 | 0.440 | 0.445 |
| | KAN | **0.310** | **0.356** | **0.350** | **0.384** | **0.368** | **0.401** | **0.433** | **0.440** |
| ETTm2 | MLP | 0.177 | 0.268 | 0.253 | 0.338 | 0.321 | 0.363 | 0.418 | 0.417 |
| | KAN | **0.176** | **0.266** | **0.249** | **0.321** | **0.314** | **0.356** | **0.410** | 0.418 |
| Exchange | MLP | 0.090 | 0.210 | 0.200 | 0.311 | 0.401 | 0.457 | 1.185 | 0.815 |
| | KAN | **0.081** | **0.200** | **0.179** | **0.300** | **0.351** | **0.424** | **0.892** | **0.708** |

## 5 CONCLUSION AND FUTURE WORK

In this paper, we introduced **TimeKAN**, a novel approach for multivariate time series forecasting that combines the strengths of Kolmogorov-Arnold Networks (KAN) with a Multi-Scale Patching (MSP) module. Our TimeKAN addresses the critical challenges of interpretability and parameter efficiency while maintaining high predictive accuracy. By leveraging KAN's ability to symbolize parameters and our innovative MSP module, TimeKAN effectively captures both temporal and cross-dimensional features across various scales. Extensive experiments conducted on six real-world datasets demonstrate that TimeKAN consistently outperforms state-of-the-art models, achieving superior performance in long-term forecasting tasks. Additionally, our TimeKAN provides insights into its decision-making process through symbolic feature extraction, enhancing transparency and interpretability.

Future work will focus on further refining the MSP module to handle even more complex temporal patterns and exploring the application of TimeKAN in other domains that require high interpretability, such as healthcare and finance. Moreover, enhancing training speed via parallel processing, effective batch management, and refined spline calculations will be essential for KAN's practical application in real-time scenarios.

## 6 REPRODUCIBILITY STATEMENT

By detailing the datasets, evaluation metrics, baseline models, and implementation configurations (more details are showed in Appendix), we aim to provide clarity and facilitate reproducibility in future research. To ensure the reproducibility of our results, we:

- Set consistent random seeds for initialization across all experiments.
- Document our experimental settings thoroughly, including data preprocessing steps, model configurations, and training procedures.
- Make all code, data splits, and configurations publicly available upon publication to facilitate future research and benchmarking. Our working code has been open sourced anonymously via Github at the URL: `https://anonymous.4open.science/r/TimeKAN-1BCD/`.

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

## A  APPENDIX

### A.1  DATASETS

To comprehensively assess the effectiveness and generalization capability of our proposed TimeKAN, we conduct experiments on multiple widely used time series datasets across different domains, including energy, meteorology, and finance. By utilizing datasets with varying characteristics and forecasting horizons, we aim to demonstrate the versatility and robustness of our TimeKAN.

Table 4: Statistics of the datasets used in experiments.

| Dataset | Domain | Variables | Samples |
|---|---|---|---|
| ETT h1/h2 | Energy | 7 | 17,420 |
| ETT m1/m2 | Energy | 7 | 69,680 |
| Weather | Meteorology | 21 | 52,696 |
| Exchange | Finance | 8 | 7,588 |
| Exchange Pro | Finance | 21 | 6284 |

**Electricity Transformer Temperature** (ETT) (Zhou et al., 2021) is critical for power system analysis and forecasting, as it records oil temperature and load data from electricity transformers. Furthermore, it includes subsets that enable evaluation on both short-term and long-term forecasting tasks with varying temporal resolutions.

**Weather** (Nie et al., 2022) contains meteorological data such as temperature, humidity, wind speed, and precipitation collected from multiple weather stations over one year. This dataset is suitable for weather forecasting tasks and tests models' abilities to handle complex seasonal patterns and multivariate dependencies.

**Exchange** (Lai et al., 2018) records the daily exchange rates of eight countries including the USA, Canada, the UK, Japan, Switzerland, Australia, and New Zealand from 1990 to 2016. It is widely used for financial forecasting and economic analysis, presenting challenges due to the inherent volatility, non-stationarity, and interdependencies in financial time series.

**Exchange Pro** (Exg Pro) is a new dataset constructed to further evaluate our TimeKAN's performance in more diverse economic contexts. This dataset encompasses foreign exchange data from **21 different economies** with varying levels of development and economic systems over a span of approximately 7,000 days. By including countries with different economic indicators and market behaviors, we aim to assess the model's capability to generalize across heterogeneous financial data and capture complex, nonlinear relationships in a broader economic.

### A.2  IMPLEMENTATION DETAILS

**Hardware and Software Configuration:** Experiments were executed using PyTorch 2.0.0 with CUDA 11.7 on a system featuring **NVIDIA RTX 3090 GPUs** (24GB VRAM) and an AMD EPYC 7352 24-Core Processor. Calculations utilized FP32 precision.

**Optimizer and Learning Rate:** We employed the **AdamW optimizer** (Loshchilov, 2017), which integrates the adaptive learning rate of Adam with decoupled weight decay. The starting learning rate was $1 \times 10^{-3}$, adjusted through a cosine annealing schedule to improve convergence.

**Batch Size and Sequence Length:** The batch size was set to 32. For long-term sequence prediction, we used an input length of 96 and prediction horizons of 96, 192, 336, and 720.

**Early Stopping:** Training ceased if validation loss did not drop for **10 consecutive epochs** to prevent overfitting and save computation. The efficient architecture of our **TimeKAN** model typically required fewer epochs due to faster convergence.

**Hyperparameter Setting:** Key hyperparameters were tuned to enhance model performance across datasets. These included hidden layer sizes in the Encoder/Decoder modules (Channel-wise KAN, Inter-patch KAN, Intra-patch KAN), number of residual-connected layers, and number of TimeKAN

blocks. Patch sizes in TimeKANs were determined using FFT to extract high-frequency periods, which were then randomly combined.

## A.3 INTERACTIVE INTERFACE FOR INTERPRETING TIMEKAN

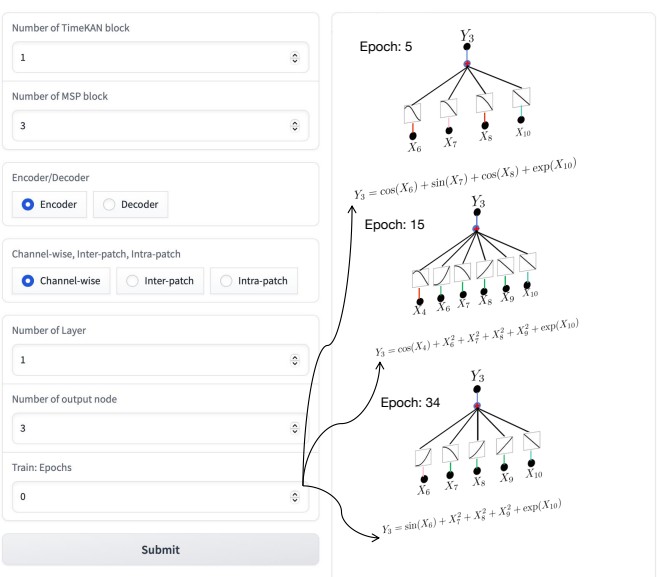

Figure 6: These images are taken from our visualization and interactive interface. To highlight the intuitiveness and interpretability of our interface, we have captured the internal structure within a KANLayer during a complete training process.

To further enhance the interpretability and transparency of our TimeKAN, we have developed an interactive interface, as shown in Figure 6. This interface allows users, prior to the commencement of training, to choose whether to apply symbolic processing and to select parameters such as the TimeKAN block number to be symbolized, the index of the MSP block within each block, the type of Encoder/Decoder module, the number of KANLayers, and the number of output nodes.

This interface is designed to provide users with "a sharp scalpel," enabling them to conveniently and efficiently dissect the internal mechanisms of the model. It allows users to observe how the model "thinks, decides, and acts," thereby mitigating, to some extent, the limitations of previous black-box neural networks in tasks that require interpretability.

## A.4 HYPERPARAMETER SETTING

In order to ensure the reproducibility of our proposed method, we have listed in the tables 5,6 below the parameters that achieved optimal performance on two metrics across multiple benchmarks. The parameters hid_chn, hid_len, hid_pch, and hid_pred represent the hidden layer sizes of the channel-wise module, intra-patch module, inter-patch module, and forecast module, respectively. The patch size refers to the suggested patch sizes obtained after preprocessing the data. KAN_Layers denotes the number of layers within each KAN-related module. Msp_num indicates the number of Msp blocks within each TimeKAN block, while Patch_num refers to the number of sequentially connected TimeKAN blocks.

Table 5: The table is the config of the hyperparameters when Our proposed method had the bset performance in MSE of each benchmarks.

| Config | | hid_chn | hid_len | hid_pch | hid_pred | patch sizes | KAN_Layers | Msp_num | Patch_num |
|---|---|---|---|---|---|---|---|---|---|
| ETTm1 | 96 | 4 | 4 | 8 | 512 | [56,42,12,6,1] | 5 | 4 | 5 |
| | 192 | 2 | 4 | 12 | 512 | [56,42,12,6,1] | 3 | 4 | 4 |
| | 336 | 1 | 1 | 2 | 512 | [56,42,12,6,1] | 3 | 1 | 5 |
| | 720 | 4 | 4 | 8 | 512 | [56,42,12,6,1] | 5 | 4 | 5 |
| ETTm2 | 96 | 2 | 2 | 8 | 512 | [56,26,12,6,1] | 3 | 4 | 5 |
| | 192 | 1 | 1 | 8 | 512 | [56,26,12,6,1] | 3 | 2 | 5 |
| | 336 | 1 | 1 | 8 | 512 | [56,26,12,6,1] | 3 | 1 | 4 |
| | 720 | 1 | 1 | 4 | 512 | [56,26,12,6,1] | 3 | 1 | 4 |
| ETTh1 | 96 | 1 | 1 | 4 | 512 | [56,42,26,12,6,1] | 3 | 1 | 6 |
| | 192 | 1 | 8 | 4 | 512 | [56,42,26,12,6,1] | 3 | 5 | 6 |
| | 336 | 2 | 8 | 8 | 512 | [56,42,26,12,6,1] | 3 | 5 | 3 |
| | 720 | 1 | 2 | 12 | 512 | [56,42,26,12,6,1] | 3 | 4 | 3 |
| ETTh2 | 96 | 2 | 8 | 8 | 512 | [58,42,14,7,1,1] | 3 | 5 | 6 |
| | 192 | 2 | 4 | 16 | 512 | [58,42,14,7,1,1] | 3 | 1 | 5 |
| | 336 | 1 | 1 | 12 | 512 | [58,42,14,7,1,1] | 3 | 4 | 6 |
| | 720 | 1 | 2 | 12 | 512 | [58,42,14,7,1,1] | 4 | 3 | 3 |
| Exchange | 96 | 8 | 8 | 16 | 256 | [59,26,11,7,2,1] | 3 | 5 | 5 |
| | 192 | 2 | 1 | 16 | 256 | [59,26,11,7,2,1] | 4 | 2 | 3 |
| | 336 | 1 | 1 | 12 | 256 | [59,26,11,7,2,1] | 3 | 3 | 3 |
| | 720 | 1 | 1 | 16 | 256 | [59,26,11,7,2,1] | 2 | 6 | 1 |
| Weather | 96 | 2 | 2 | 8 | 256 | [26,11,7,2,1,1] | 3 | 1 | 3 |
| | 192 | 1 | 8 | 12 | 256 | [26,11,7,2,1,1] | 3 | 2 | 4 |
| | 336 | 1 | 4 | 4 | 256 | [26,11,7,2,1,1] | 3 | 3 | 6 |
| | 720 | 1 | 2 | 12 | 256 | [26,11,7,2,1,1] | 3 | 2 | 4 |
| Exg Pro | 96 | 1 | 1 | 2 | 256 | [59,26,11,7,2,1] | 3 | 4 | 4 |
| | 192 | 1 | 1 | 12 | 256 | [59,26,11,7,2,1] | 3 | 5 | 4 |
| | 336 | 2 | 8 | 2 | 256 | [59,26,11,7,2,1] | 3 | 2 | 6 |
| | 720 | - | - | - | - | - | - | - | - |

Table 6: The table is the config of the hyperparameters when Our proposed method had the best performance in MAE of each benchmark.

| Config | | hid_chn | hid_len | hid_pch | hid_pred | patch sizes | KAN_Layers | Msp_num | Patch_num |
|---|---|---|---|---|---|---|---|---|---|
| ETTm1 | 96 | 4 | 4 | 8 | 512 | [56,42,12,6,1] | 5 | 4 | 5 |
| | 192 | 2 | 4 | 12 | 512 | [56,42,12,6,1] | 3 | 4 | 4 |
| | 336 | 1 | 1 | 2 | 512 | [56,42,12,6,1] | 3 | 1 | 5 |
| | 720 | 2 | 4 | 12 | 512 | [56,42,12,6,1] | 3 | 1 | 5 |
| ETTm2 | 96 | 2 | 2 | 8 | 512 | [56,26,12,6,1] | 3 | 4 | 5 |
| | 192 | 1 | 2 | 4 | 512 | [56,26,12,6,1] | 3 | 5 | 4 |
| | 336 | 1 | 1 | 8 | 512 | [56,26,12,6,1] | 3 | 1 | 4 |
| | 720 | 1 | 1 | 4 | 512 | [56,26,12,6,1] | 3 | 1 | 4 |
| ETTh1 | 96 | 1 | 1 | 4 | 512 | [56,42,26,12,6,1] | 3 | 1 | 6 |
| | 192 | 1 | 2 | 4 | 512 | [56,42,26,12,6,1] | 3 | 5 | 3 |
| | 336 | 2 | 4 | 2 | 512 | [56,42,26,12,6,1] | 3 | 1 | 6 |
| | 720 | 1 | 4 | 12 | 512 | [56,42,26,12,6,1] | 3 | 4 | 3 |
| ETTh2 | 96 | 1 | 1 | 4 | 512 | [58,42,14,7,1,1] | 3 | 1 | 6 |
| | 192 | 2 | 4 | 16 | 512 | [58,42,14,7,1,1] | 3 | 1 | 5 |
| | 336 | 1 | 1 | 12 | 512 | [96,48,18,6,2,1] | 3 | 4 | 5 |
| | 720 | 1 | 2 | 16 | 512 | [58,42,14,7,1,1] | 4 | 1 | 4 |
| Exchange | 96 | 8 | 8 | 16 | 256 | [59,26,11,7,2,1] | 3 | 5 | 5 |
| | 192 | 4 | 1 | 16 | 256 | [59,26,11,7,2,1] | 4 | 2 | 4 |
| | 336 | 1 | 1 | 12 | 256 | [59,26,11,7,2,1] | 3 | 3 | 3 |
| | 720 | 2 | 2 | 16 | 256 | [59,26,11,7,2,1] | 4 | 3 | 1 |
| Weather | 96 | 2 | 2 | 8 | 256 | [26,11,7,2,1,1] | 3 | 1 | 3 |
| | 192 | 1 | 8 | 12 | 256 | [26,11,7,2,1,1] | 3 | 2 | 4 |
| | 336 | 1 | 4 | 4 | 256 | [26,11,7,2,1,1] | 3 | 3 | 6 |
| | 720 | 1 | 2 | 12 | 256 | [26,11,7,2,1,1] | 3 | 2 | 4 |
| Exg Pro | 96 | 1 | 1 | 2 | 256 | [59,26,11,7,2,1] | 3 | 4 | 4 |
| | 192 | 1 | 1 | 12 | 256 | [59,26,11,7,2,1] | 3 | 5 | 4 |
| | 336 | 2 | 8 | 2 | 256 | [59,26,11,7,2,1] | 3 | 2 | 6 |
| | 720 | - | - | - | - | - | - | - | - |

## A.5 ABLATION STUDY

Additionally, to demonstrate how the proposed MSP strategy benefits MTS forecasting, we conducted ablation experiments on the number of MSP blocks within the TimeKAN block. The results are as follows 7.

Table 7: Performance of our proposed method under different MSP parameters across various time series benchmark datasets. Except for the MSP parameters, all other parameters are set to our TimeKAN's optimal parameters for each corresponding benchmark, which are listed above.

| # MSP | | 1 | | 2 | | 3 | | 4 | | 5 | |
|---|---|---|---|---|---|---|---|---|---|---|---|
| Metric | | MSE | MAE | MSE | MAE | MSE | MAE | MSE | MAE | MSE | MAE |
| ETTm1 | 96 | 0.335 | 0.379 | 0.320 | 0.371 | 0.317 | 0.368 | 0.310 | 0.356 | 0.319 | 0.374 |
| | 192 | 0.365 | 0.397 | 0.356 | 0.391 | 0.351 | 0.386 | 0.350 | 0.384 | 0.354 | 0.361 |
| | 336 | 0.386 | 0.403 | 0.376 | 0.403 | 0.375 | 0.407 | 0.368 | 0.401 | 0.387 | 0.413 |
| | 720 | 0.448 | 0.445 | 0.445 | 0.453 | 0.451 | 0.457 | 0.433 | 0.440 | 0.447 | 0.453 |
| ETTm2 | 96 | 0.176 | 0.266 | 0.190 | 0.275 | 0.187 | 0.271 | 0.254 | 0.313 | 0.211 | 0.292 |
| | 192 | 0.247 | 0.312 | 0.249 | 0.321 | 0.243 | 0.309 | 0.254 | 0.317 | 0.247 | 0.345 |
| | 336 | 0.314 | 0.356 | 0.385 | 0.416 | 0.352 | 0.388 | 0.374 | 0.391 | 0.341 | 0.347 |
| | 720 | 0.410 | 0.418 | 0.444 | 0.447 | 0.429 | 0.433 | 0.437 | 0.451 | 0.431 | 0.443 |
| ETTh1 | 96 | 0.376 | 0.396 | 0.374 | 0.399 | 0.375 | 0.402 | 0.378 | 0.400 | 0.382 | 0.403 |
| | 192 | 0.428 | 0.435 | 0.429 | 0.443 | 0.424 | 0.435 | 0.427 | 0.439 | 0.419 | 0.426 |
| | 336 | 0.469 | 0.467 | 0.464 | 0.466 | 0.459 | 0.463 | 0.469 | 0.470 | 0.449 | 0.450 |
| | 720 | 0.507 | 0.513 | 0.491 | 0.500 | 0.547 | 0.535 | 0.464 | 0.475 | 0.466 | 0.481 |
| ETTh2 | 96 | 0.286 | 0.346 | 0.292 | 0.355 | 0.289 | 0.347 | 0.298 | 0.353 | 0.282 | 0.342 |
| | 192 | 0.365 | 0.391 | 0.386 | 0.409 | 0.368 | 0.400 | 0.383 | 0.408 | 0.376 | 0.405 |
| | 336 | 0.401 | 0.423 | 0.425 | 0.440 | 0.421 | 0.436 | 0.407 | 0.427 | 0.430 | 0.451 |
| | 720 | 0.427 | 0.454 | 0.435 | 0.467 | 0.415 | 0.469 | 0.448 | 0.486 | 0.432 | 0.460 |
| Exchange | 96 | 0.100 | 0.223 | 0.089 | 0.206 | 0.089 | 0.207 | 0.101 | 0.218 | 0.082 | 0.201 |
| | 192 | 0.211 | 0.327 | 0.175 | 0.302 | 0.167 | 0.289 | 0.196 | 0.311 | 0.178 | 0.299 |
| | 336 | 0.434 | 0.474 | 0.405 | 0.462 | 0.303 | 0.399 | 0.381 | 0.448 | 0.349 | 0.431 |
| | 720 | 1.161 | 0.818 | 1.051 | 0.770 | 1.101 | 0.795 | 1.176 | 0.818 | 1.175 | 0.822 |
| Exg Pro | 96 | 0.212 | 0.284 | 0.216 | 0.284 | 0.210 | 0.280 | 0.177 | 0.267 | 0.206 | 0.281 |
| | 192 | 0.429 | 0.415 | 0.447 | 0.422 | 0.439 | 0.417 | 0.424 | 0.420 | 0.422 | 0.414 |
| | 336 | 0.717 | 0.535 | 0.694 | 0.536 | 0.695 | 0.534 | 0.906 | 0.641 | 0.669 | 0.517 |
| | 720 | - | - | - | - | - | - | - | - | - | - |

Table 8: Comparison of the performance between MLP and KAN with the same parameters except for the hidden layer parameters.

| Benchmark | ETT m1 96 | | | | ETT m1 192 | | | | ETT m1 336 | | | | ETT m1 720 | | | |
|---|---|---|---|---|---|---|---|---|---|---|---|---|---|---|---|---|
| Metric | MSE | MAE | MSE | MAE | MSE | MAE | MSE | MAE | MSE | MAE | MSE | MAE | MSE | MAE | MSE | MAE |
| | 0.310 | 0.356 | 0.313 | 0.358 | 0.350 | 0.384 | 0.367 | 0.388 | 0.368 | 0.401 | 0.374 | 0.407 | 0.433 | 0.440 | 0.440 | 0.445 |
| | ETT m2 96 | | | | ETT m2 192 | | | | ETT m2 336 | | | | ETT m2 720 | | | |
| | MSE | MAE | MSE | MAE | MSE | MAE | MSE | MAE | MSE | MAE | MSE | MAE | MSE | MAE | MSE | MAE |
| | 0.176 | 0.266 | 0.177 | 0.268 | 0.249 | 0.321 | 0.253 | 0.338 | 0.314 | 0.356 | 0.321 | 0.363 | 0.410 | 0.418 | 0.418 | 0.417 |
| | ETT h1 96 | | | | ETT h1 192 | | | | ETT h1 336 | | | | ETT h1 720 | | | |
| | MSE | MAE | MSE | MAE | MSE | MAE | MSE | MAE | MSE | MAE | MSE | MAE | MSE | MAE | MSE | MAE |
| | 0.376 | 0.395 | 0.377 | 0.397 | 0.419 | 0.426 | 0.425 | 0.428 | 0.449 | 0.450 | 0.491 | 0.459 | 0.464 | 0.475 | 0.497 | 0.485 |
| | ETT h2 96 | | | | ETT h2 192 | | | | ETT h2 336 | | | | ETT h2 720 | | | |
| | MSE | MAE | MSE | MAE | MSE | MAE | MSE | MAE | MSE | MAE | MSE | MAE | MSE | MAE | MSE | MAE |
| | 0.282 | 0.342 | 0.297 | 0.355 | 0.365 | 0.391 | 0.377 | 0.397 | 0.407 | 0.427 | 0.416 | 0.434 | 0.415 | 0.448 | 0.439 | 0.465 |
| | Exchange 96 | | | | Exchange 192 | | | | Exchange 336 | | | | Exchange 720 | | | |
| | MSE | MAE | MSE | MAE | MSE | MAE | MSE | MAE | MSE | MAE | MSE | MAE | MSE | MAE | MSE | MAE |
| | 0.081 | 0.200 | 0.090 | 0.210 | 0.179 | 0.300 | 0.200 | 0.311 | 0.351 | 0.424 | 0.401 | 0.457 | 0.892 | 0.708 | 1.185 | 0.815 |
| | Weather 96 | | | | Weather 192 | | | | Weather 336 | | | | Weather 720 | | | |
| | MSE | MAE | MSE | MAE | MSE | MAE | MSE | MAE | MSE | MAE | MSE | MAE | MSE | MAE | MSE | MAE |
| | 0.153 | 0.217 | 0.161 | 0.222 | 0.201 | 0.267 | 0.205 | 0.276 | 0.256 | 0.310 | 0.267 | 0.319 | 0.326 | 0.361 | 0.338 | 0.364 |
| | Exchange Pro 96 | | | | Exchange Pro 192 | | | | Exchange Pro 336 | | | | Exchange Pro 720 | | | |
| | MSE | MAE | MSE | MAE | MSE | MAE | MSE | MAE | MSE | MAE | MSE | MAE | MSE | MAE | MSE | MAE |
| | 0.177 | 0.257 | 0.181 | 0.262 | 0.338 | 0.362 | 0.352 | 0.368 | 0.589 | 0.485 | 0.603 | 0.498 | - | - | - | - |

Additionally, in our work, we employed an emerging network structure, KAN. To verify the effectiveness of KAN in our task, we replaced [1, 2, 4, 8, 12, 16] with [16, 32, 64, 128, 256, 512], and replaced KAN with MLP in the structure. The validation results of our experiments are presented in the table below 8.

