# OpenReview forum: "TimeKAN: A Transparent KAN-Based Approach for Multivariate Time Series Forecasting"
_ICLR.cc/2025/Conference — ICLR 2025 Conference Withdrawn Submission_

### Official Review · Reviewer_X5XU · 2024-10-15

**Soundness:** 3
**Presentation:** 3
**Contribution:** 3
**Rating:** 6
**Confidence:** 3

**Summary:**

The paper titled "TimeKAN: A Transparent KAN based Multivariate Time Series Prediction Method" introduces TimeKAN, a new method for multivariate time series forecasting (MTSF) using Kolmogorov Arnold network (KAN). TimeKAN addresses challenges such as lack of interpretability and difficulty in capturing long-term dependencies and cross dimensional relationships in MTSF. This model introduces a Multi Scale Splicing (MSP) module to capture features at different temporal resolutions, while using symbolic techniques to enhance interpretability by visualizing the features. Extensive experiments on six real-world datasets have shown that TimeKAN outperforms state-of-the-art models in both prediction accuracy and interpretability, making it highly suitable for complex time series tasks.

**Strengths:**

Originality: In the context of multivariate time series prediction, using Kolmogorov Arnold network (KAN) to bridge the gap between interpretability and accuracy is a new approach. KAN allows for symbolic representation of learned features, which improves the transparency of the model compared to deep learning models, especially in Transformer based architectures that often lack interpretability.

Quality: The design of the MSP module is particularly noteworthy as it captures temporal dependencies at different scales. By dividing the input sequence into patches of different sizes, this model can capture short-term and long-term dependencies. The experimental section clearly demonstrated that the model consistently outperforms state-of-the-art models on six benchmark datasets, including ETT, Weather, and Exchange.

Clarity: This article provides a clear explanation of the architecture innovation of TimeKAN, particularly the detailed decomposition of multi-scale patching mechanisms (pages 6-7) and the symbolic regression technique used to explain learning representations. The addition of visualization tools further increases the transparency of the model.

**Weaknesses:**

Computational efficiency: Although TimeKAN has strong predictive accuracy, its computational efficiency on very large datasets or ultra long sequences may become a bottleneck. The MSP module increases the complexity of the model by generating multiple patch sizes, which may result in high computational costs, especially when applied to datasets outside of those included in the experiment. It would be beneficial to discuss in more detail the potential of parallelization or distributed training to alleviate this situation.

Extension to other fields: Although the datasets used for evaluation are diverse, they are limited to specific fields such as energy, meteorology, and finance. There is no detailed exploration of extending symbolic feature representation to other fields, such as biological or industrial time series data. Future work may include verifying the interpretability of TimeKAN in more diverse fields.

**Questions:**

Computational complexity: Considering the potential overhead introduced by the MSP module, could you discuss any optimizations you are considering to improve the performance of the model on large datasets or ultra long sequences?

Symbolic representation generalization: Have you tested symbolic feature interpretation in other types of data, such as biological or industrial datasets? How common is this interpretable feature in areas outside of your experiment?

Long term prediction: How will TimeKAN's performance decline in very long-term predictions? For example, does the model maintain accuracy in predicting the next few weeks, and in these cases, how does it compare to simpler models such as DLlinear?

---

### Official Review · Reviewer_U1Xw · 2024-10-28

**Soundness:** 4
**Presentation:** 3
**Contribution:** 3
**Rating:** 3
**Confidence:** 4

**Summary:**

The paper introduces TimeKAN, a novel approach for multivariate time series forecasting that leverages Kolmogorov-Arnold Networks (KAN) and a Multi-Scale Patching (MSP) module. TimeKAN addresses the challenges of interpretability and parameter efficiency while maintaining high predictive accuracy. The model captures both temporal and cross-dimensional features across various scales and provides insights into its decision-making process through symbolic feature extraction.

**Strengths:**

1. TimeKAN enhances model interpretability through symbolic regression, allowing for the extraction of human-readable models that explain underlying data patterns. The model achieves comparable accuracy to MLPs with significantly fewer parameters, which is beneficial for computational efficiency and model training.
2. The MSP module effectively captures multi-period dependencies between variates, allowing the model to focus on a broader spectrum of temporal patterns. And TimeKAN outperforms existing methods, including both MLP-based and Transformer-based architectures, across various datasets and forecasting horizons.

**Weaknesses:**

1. TimeKAN may not be the best performer in ETTm2, as indicated by MSD-Mixer outperforming it for every output length. The effectiveness of the MSP module relies on the determination of patching sizes through FFT, which may introduce complexity in data preprocessing.
2. The paper primarily focuses on performance on seen datasets, and it is unclear how TimeKAN would generalize to unseen or significantly different data distributions. While the model is parameter-efficient, the computational cost of training and inference, especially for real-time scenarios, is not explicitly addressed.
3. The paper could benefit from a more detailed discussion on scenarios where TimeKAN may underperform or fail, which would aid in understanding its limitations.

**Questions:**

As is commented in Weakness.

---

### Official Review · Reviewer_3zmk · 2024-10-31

**Soundness:** 2
**Presentation:** 2
**Contribution:** 2
**Rating:** 3
**Confidence:** 4

**Summary:**

The article proposes TimeKAN, a model based on Kolmogorov-Arnold Networks for multivariate time series forecasting. TimeKan employs symbolization techniques to represent the learned features and visualize the training process, addressing the interpretability challenge. Experimentally, TimeKAN outperforms state-of-the-art (SOTA) methods in terms of predictive performance.

**Strengths:**

1. This paper introduces Kolmogorov-Arnold Networks to time series data to improve the performance and interpretability.
2. The method is simple and intuitively effective.

**Weaknesses:**

1. The overall novelty is limited. Since the hierarchical decomposition architectures have already been studied by N-BEATS and N-HiTs, the patch size selecting method is similar to the existing TimesNet.
2. Although the authors have included some MLP-based as baselines, some highly-related works is omitted, such as N-BEATS [1] and Pathformer [2].
[1] Oreshkin, Boris N., et al. "N-BEATS: Neural basis expansion analysis for interpretable time series forecasting." ICLR 2020.
[2] Chen, Peng, et al. "Pathformer: Multi-scale transformers with adaptive pathways for time series forecasting." ICLR 2024.
3. Since the author argues that existing MLP-based and Transformer-based models lack interpretability due to their large parameter sizes (Line 15), and propose the effective TimeKAN, detailed efficiency analysis is required.

**Questions:**

1. Could you give more explanation on the notations $x_1, ..., x_5$ in Figure 1 and $X_6, ..., X_{10}$ in Figure 5?
2. In the experiments, why did you choose not to use ECL and Traffic datasets? They are also well-established multivariate forecasting benchmarks similar to ETT.
3. Could you provide more discussion on the experimental results in Table 2? Why did the number of MSPs introduce such a large fluctuation in the results?
4. Could you provide some intuitive showcase of the prediction results?

---

### Official Review · Reviewer_EkiL · 2024-11-02

**Soundness:** 3
**Presentation:** 3
**Contribution:** 2
**Rating:** 3
**Confidence:** 4

**Summary:**

In this paper, TimeKAN, a Kolmogorov-Arnold network based time series model, is proposed and demonstrate state-of-the-art prediction performance on some real-world datasets.

**Strengths:**

1. This paper attempts to introduce the KAN and a Multi-Scale Patching module  for time series forecasting, aiming to achieve lower model parameters and better prediction performance.

2. Try to explore the predictive and interpretable capabilities of the proposed method in time series.​

**Weaknesses:**

​Lack of comprehensive explanation and analysis of the benefits of introducing new architectures in place of well-used model modules and incomplete experimental validation.​

**Questions:**

1. ​The proposed method lacks novelty and introduces a new architecture in time series prediction, but there is insufficient theoretical justification for the benefits of introducing KAN into time series prediction.​

2. The paper emphasizes that KAN has fewer parameters compared to MLP, yet it lacks a comparative analysis of parameter counts and effectiveness against existing MLP-based models such as Dlinear, Nbeats, RLinear, and Transformer-based models.

3. Experimental comparisons with baselines are not fully fair, as they lack tests on multidimensional datasets such as ECL and Traffic, which include more channels and larger data sizes.  Some advanced time series models such as iTransformer can better accommodate these prediction scenarios.

4. Ablation studies are incomplete and require a thorough comparison across all datasets, rather than just some small datasets ETT and Exchange. Moreover, the improvement from replacing MLP with KAN on existing datasets is not significant. This conclusion can be seen by the relative improvement in prediction performance.​

---

### Note · Authors · 2024-11-18

I have read and agree with the venue's withdrawal policy on behalf of myself and my co-authors.